

# First *ex situ* outplanting of the habitat-forming seaweed *Cystoseira amentacea* var. *stricta* from a restoration perspective

Gina De La Fuente[1], Mariachiara Chiantore[1], Valentina Asnaghi[1], Sara Kaleb[2] and Annalisa Falace[2]

[1] Department of Earth, Environment and Life Sciences, University of Genoa, Genoa, Italy
[2] Department of Life Sciences, University of Trieste, Trieste, Italy

## ABSTRACT

In the Mediterranean Sea, brown algae belonging to the *Cystoseira* genus play a valuable role as foundation species. Due to evidences of regression/loss of the habitats of these species caused by the interplay of human and climatic disturbances, active restoration measures have been encouraged by EU regulations. In particular, nondestructive restoration techniques, which avoid the depletion of threatened species in donor populations, are strongly recommended. In the framework of the EU project ROCPOP-Life, the first *ex situ* outplanting experience of *Cystoseira amentacea* var. *stricta* has been implemented in the Cinque Terre Marine Protected Area (northwestern Mediterranean). A total of 400 clay tiles, hosting approximately three mm-long germlings of *C. amentacea*, were fixed to the rocky shore with screws: the tiles were monitored for the next 2 months by photographic sampling, and survival (presence/absence of juveniles on the tiles), cover and growth were assessed. Additional sampling was performed 6 months after tile deployment, after which an unprecedented storm surge severely affected the restoration performance. After 2 months, over 40% of the tiles were covered with *Cystoseira* juveniles, which reached approximately eight mm in total length. The tiles that survived the storm hosted three to six cm-long juveniles. The high cover ($\geq$25%), assuring moisture and shading, and the appropriate size of the juveniles, to avert micro-grazing, at time of deployment were key to the survival and growth of the outplanted juveniles, increasing the potential for restoration success. Our findings show that outplanting of midlittoral canopy-forming species is a feasible approach for restoration efforts, with particular attention given to the early phases: (i) laboratory culture, (ii) transport, and (iii) juvenile densities. These results are strongly encouraging for the implementation of restoration actions for *C. amentacea* on a large scale, in light of EU guidelines.

# INTRODUCTION

Approximately 35% of brown algae species (Laminariales and Fucales; *Guiry & Guiry, 2019*) play a key role as ecosystem engineers, building kelp forests, and fucoid canopies,

Corresponding author
Gina De La Fuente,
gina.delafuente@edu.unige.it

which enhance habitat complexity, biodiversity, ecosystem functioning, and the natural capital of littoral ecosystems (*Thompson et al., 1996*; *Christie, Jørgensen & Norderhaug, 2007*; *Airoldi et al., 2014*). Kelp forests (i.e., *Macrocystis*, *Lessonia*, and *Laminaria*) constitute one of the most diverse and productive ecosystems distributed worldwide (*Steneck et al., 2002*). In the Mediterranean Sea, some Laminariales (e.g., *Laminaria* and *Sacchoriza*) and Fucales (e.g., *Sargassum* and *Cystoseira*) also play a role as foundation species in some specific locations, but the canopy-forming brown algae of the *Cystoseira* genus are the most important because they are widespread in this biogeographic region (*Rodríguez-Prieto et al., 2013*). However, they are exposed to multiple disturbances that cause a decline in their abundance in many coastal areas (*Airoldi, Balata & Beck, 2008*; *Mineur et al., 2015*). The main pressures affecting the valuable ecosystems formed by *Cystoseira* are sedimentation (*Perkol-Finkel & Airoldi, 2010*), low water quality (*Arévalo, Pinedo & Ballesteros, 2007*; *Sales et al., 2011*), anthropization (*Mangialajo, Chiantore & Cattaneo-Vietti, 2008*), and overgrazing (*Sala, Boudouresque & Harmelin-Vivien, 1998*; *Hereu, 2004*; *Verges et al., 2014*).

Several studies conducted in the Mediterranean have reported on the past and the present distribution and abundance of *Cystoseria* canopies (*Thibaut et al., 2014*; *Mancuso et al., 2018*), detecting regressions or losses caused by the above mentioned factors (*Cormaci & Furnari, 1999*; *Thibaut et al., 2005*; *Falace et al., 2010*; *De La Fuente et al., 2018*). Their natural recovery, in the absence of adults, is hampered by the very limited dispersal of *Cystoseira* species due to the rapid fertilization of their large eggs and zygote sinking (*Falace et al., 2018*).

In this framework, active marine restoration is strongly encouraged to prevent the loss of the valuable habitats formed by *Cystoseira* species that enhance biodiversity and preserve ecosystem functions and services, according to the Biodiversity Strategy to 2020 (Target 2; *European Commission, 2011*).

Three different restoration techniques have been implemented in the Mediterranean Sea for *Cystoseira* species: (i) transplanting juveniles or adults (*Falace, Zanelli & Bressan, 2006*; *Susini et al., 2007*), (ii) positioning fertile receptacles in the target area (*in situ*; *Verdura et al., 2018*), or (iii) outplanting juveniles cultured in the laboratory along the shore (*ex situ*; *Sales et al., 2011*; *Verdura et al., 2018*). The latter two techniques are strongly recommended for the restoration of threatened species to avoid the depletion of natural donor populations (*Falace et al., 2018*). Active restoration actions must be implemented depending on the biological traits of the target species (i.e., reproductive strategy) and the environmental characteristics (i.e., depth and hydrodynamics). The *in situ* technique seems to be especially suitable for species with high dispersal capacity (e.g., kelps; *Reed, Laur & Ebeling, 1988*; *Gaylord et al., 2002*), while the *ex situ* technique is more appropriate for species with a low dispersal capacity (e.g., *Cystoseira amentacea*; *Mangialajo et al., 2012*). Both techniques have been recently validated for a shallow species (*C. barbata*; *Verdura et al., 2018*) living in low hydrodynamic conditions, but at present, such approaches have never been tested for the midlittoral habitat.

The aim of this study was to apply the *ex situ* technique (outplanting) to midlittoral *C. amentacea* var. *stricta* Montagne (hereafter *C. amentacea*), in the Ligurian Sea

(northwestern Mediterranean), assessing its survival, cover and growth in the first, most critical, months following implementation. The outplanting was performed in the Cinque Terre Marine Protected Area (Cinque Terre MPA), where, at present, only the more tolerant congener *C. compressa* is found in continuous and discontinuous belts (*Asnaghi et al., 2009*; *De La Fuente et al., 2018*). Here, *C. amentacea* is presently missing as well as across the whole easternmost side of the Ligurian coastline starting from Punta Manara, which is approximately 20 km northwest of the Cinque Terre MPA (author's personal observation). This species was recorded in the area until the end of the 19th century (authors' personal observation through herbarium records, *De La Fuente et al., 2018*) and its disappearance can be explained by habitat fragmentation due to coastal developments, water pollution and high sediment loads. Such stressors were related to the intense excavating activities that were carried out to extract construction material (mostly for building railways and highways in the area), which occurred in the first half of the 20th century. Even though such disruptive activities have largely been reduced in recent decades, with significant changes in the riverine basin and sediment load to the sea, and a MPA was established (1997), this sensitive species is still absent along these stretches of coastline. This is likely due to its above mentioned low dispersal capacity (<1 m, *Johnson & Brawley, 1998*; *Mangialajo et al., 2012*) which hampers the natural recovery of this species, in addition to other local factors, such as mussel farming (as a result of competition for space with the settled mussel juveniles).

The effectiveness of the *ex situ* approach over the first 6 months following its implementation, was assessed in terms of the presence, cover and growth of the outplanted *C. amentacea* juveniles in the framework of the EU project ROCPOP-Life.

## MATERIALS AND METHODS

### Study sites

The *ex situ* outplanting of the midlittoral species *C. amentacea* was performed in summer 2018 following a nondestructive strategy, with apical fronds (ca. three cm in length) holding mature receptacles collected in June from a healthy population located in the Portofino MPA (donor site) along a 200-m stretch of coastline. After a laboratory culturing period, the juveniles were transplanted into the Cinque Terre MPA (receiving site; approximately 80 km from the donor site). Both MPAs are located in the Ligurian Sea, northwestern Mediterranean (Fig. 1). The sites are characterized by a tide in the range of 30 cm (under these conditions, barometric pressure effects may have a dominant effect on the water level) and an average spring temperature of 20 °C. After sampling, the apices were gently cleaned with tweezers and rinsed with filtered seawater to remove adherent biofouling and detritus. Then, the apices, which were wrapped in seawater-wetted towels, were delivered within 24 h to the laboratory in Trieste (northeastern Italy; Fig. 1) under dark, cold, and humid conditions for culturing in environmentally controlled rooms.

### Laboratory *ex situ* cultivation

Three apices with mature receptacles (additionally cleaned with a brush and rinsed with autoclaved seawater) were placed on each substrate constituted of a rough round tiles

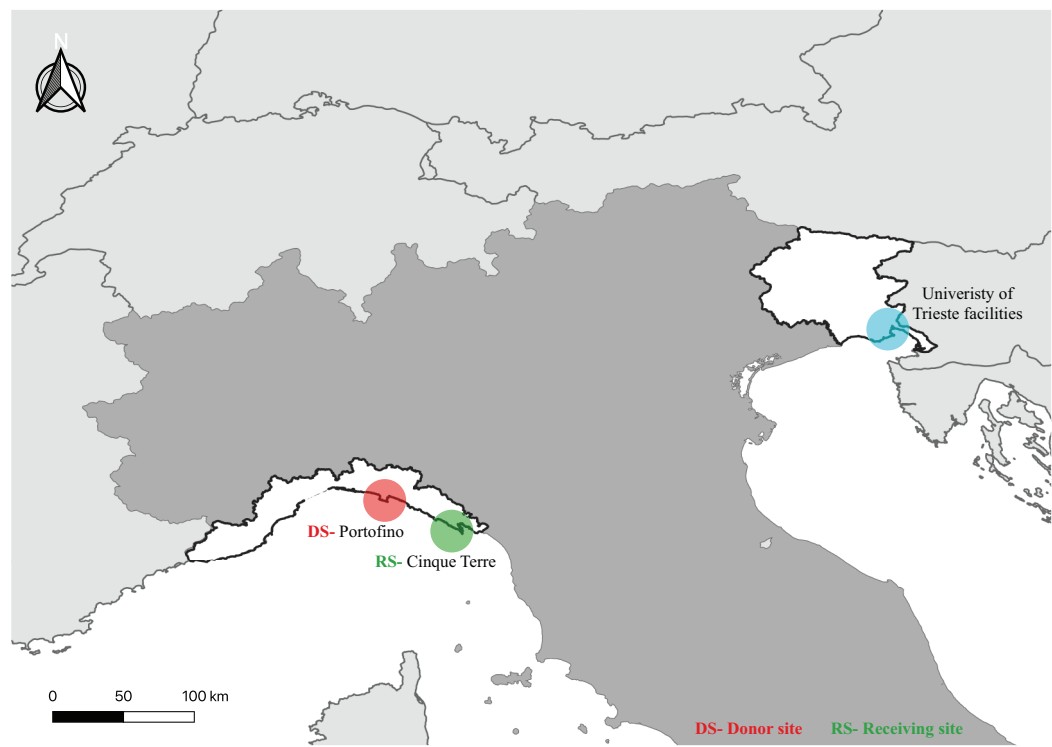

**Figure 1 Map showing the location of the culture laboratory facilities and the donor (DS) and receiving (RS) sites.**

composed of clay (4.5 cm in diameter) with a 0.6 cm hole at the center to fix the tile onto the rocks with a screw. On the next day, the fertile apices were removed, and the zygotes attached to the tiles were cultured over a 3-week period (≈450 tiles).

Temperature (+20 °C) and photoperiod (15:9 h light:dark cycle) and light intensity (125 μmoL photons m$^{-2}$ s$^{-1}$) were selected according to the protocol provided in *Falace et al. (2018)*. Von Stosch's enriched filtered seawater was used as the culture medium to accelerate the culturing time for the *C. amentacea* individuals to reach a greater size at the time of outplanting (*Falace et al., 2018*). The medium was enriched with antibiotic (two μL of amikacin sulfate and 500 μg of ampicillin sodium L$^{-1}$ of culture medium) and GeO$_2$ (*Falace, Zanelli & Bressan, 2006*) to prevent bacterial and diatom growth. The culture medium in the aquaria was renewed every 3 days to minimize any possible effects of nutrient limitation and was continuously aerated by bubbling and water multifunction pumps (≈300 L h$^{-1}$ flow) to increase oxygenation and hydrodynamics.

After 24 days of controlled growth in the Trieste laboratory, photographs were taken (17th July; 160 random tiles) in order to assess the percent cover and juvenile length.

## Outplanting and monitoring in the field

On 19th July 2018 the tiles were transported to Cinque Terre MPA. All the tiles were carefully placed in small boxes filled with filtered seawater, which were placed in a large insulated container that was maintained at a cool temperature with icepacks. The container was transported by car with an air conditioner (≈7-h trip, temperature

in the range of 20–22 °C) to the receiving site, where the boxes were stored in an air-conditioned room overnight (at 22 °C).

On the next day (20th July 2018, Time 0), the tiles were carefully transported to the field, using a rubber boat. Eight patches (Fig. S1) were established in the previous weeks: 50 holes were drilled in each patch and screws placed within them in advance. On the day of implementation, the tiles were quickly screwed to the rocks. Overall, the deployment of 400 tiles was performed in approximately 5 h. On the days of transport and deployment, the air temperature was in the range of 23–27 °C and the sea temperature was in the range of 25–26 °C.

Monitoring of the clay tiles started on the same day (Time 0) and continued over the following 2 months: July 27th (Time 1), August 6th (Time 2), August 29th (Time 3), and September 27th (Time 4). During each sampling time, photos were taken of a randomly selected 20 of the 50 clay tiles in each patch. The photos were analyzed in the lab using Image J to assess percent cover and survival (i.e., overall presence/absence of juveniles). Thallus length was measured for 40 randomly selected individual specimens with ImageJ at each sampling time except Time 5 (measured in the field).

Additional sampling was performed 4 months later (Time 5). This sampling occurred after an unprecedented storm surge that affected the Ligurian Sea at the end of October 2018 (*ANSA, 2018*; *The MediTelegraph, 2018*), which caused great destruction along the entire Ligurian coastline, with estimated damages of hundreds of millions of euros. This storm caused the destruction of entire harbors and ships in marinas, particularly along the eastern side of the Ligurian coast. Huge damage was reported among the shallow benthic communities, particularly the *Posidonia* meadows (*IL Secolo XIX, 2018*). Because most of the tiles were detached by this unprecedented event (180 km h$^{-1}$ winds and waves greater than 10 m high), it was not possible to quantitatively assess restoration performance in terms of percent cover but only in terms of juvenile growth (length) on the 16 tiles with *C. amentacea* juveniles of the approximatively 69 remaining tiles.

## Data analysis

One-way ANOVA was applied to assess possible differences in the percent cover of the juveniles on the tiles when leaving the laboratory versus at the time of the deployment (Time 0) after arcsin transformation of the data and verification of assumptions (normality using the Shapiro–Wilk test and homoscedasticity using Bartlett's test).

The effect of percent cover at the time of deployment was assessed using a generalized linear model (GLM) for both the percent cover (family = quasibinomial) and the presence/absence (family=binomial) of juveniles on the tiles at Time 4, using the "cover class" of each patch at the start of the experiment as the predictor variable. Patches were classified according to 3 "cover classes" (based on the percent cover at the start of the experiment): Low (18.2 ± 1.6, avg ± SE; patches 2, 3, and 5), Medium (25.0 ± 2.1; patches 1, 7, and 8), and High (32.2 ± 2.5; patches 4 and 6). Statistical analyses were performed with the free software R (*R Core Team, 2015*) using the stats package.

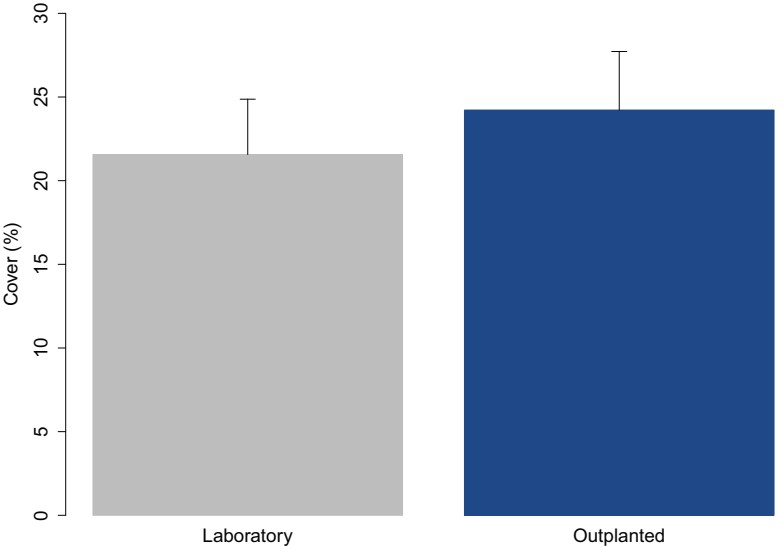

**Figure 2 Percent cover of *C. amentacea* (average + SE) on tiles when leaving the controlled growth conditions (laboratory) and at the time of positioning (outplanting) at the receiving site (72 h later).**

## RESULTS

The juveniles were 2.65 ± 0.05 mm (avg ± SE) in length when they were transported to the receiving site. The percent cover on the tiles measured at Time 0 was compared with that at the time they left the laboratory (3 days before) to assess the effect of transport (Fig. 2). The percent cover in the laboratory and in outplants at Time 0 did not significantly differ (one-way ANOVA; $p = 0.327$).

At the start of tile deployment, the average percent cover of the juveniles was 24.22 (±1.24%). The percent cover data of the juveniles on the tiles in the eight patches over time (Time 0–Time 4) are reported in Fig. 3. All patches showed a sharp decline in percent cover from Time 0 to Time 1 due to the predictable loss of some juveniles in the field. Some patches clearly showed lower average values of cover at Time 0 (ranging from 15.88 ± 2.63 in patch 2 to 32.48 ± 2.59 in patch 6; avg ± SE), and this difference in cover at the start of the experiment affected the survival and growth of the juveniles over time (Fig. 3). In the patches characterized by higher cover of juveniles at the beginning of the experiment (patches 4 and 6), after the first decline, cover increased on the following sampling dates (reaching 27.37 ± 2.63 and 11.65 ± 3.99 at Time 4 in patches 4 and 6, respectively) because of the growth of the juveniles (Figs. 3 and 4), while the patches characterized by lower cover showed a general decrease in percent cover (reaching 3.06 ± 1.55 and 4.85 ± 2.88 at Time 4 in patches 2 and 3, respectively; juveniles lost in patch 5). The GLM for the percent cover on the tiles at Time 4 showed significant differences among the classes ($p = 0.0143$). The Low class was significantly differed from the others, while the Medium class did not differ from the High class (Table 1).

The good performance of the outplanting, at least in the early phases, is confirmed by the assessment of the percentage of tiles with the presence of juveniles (Fig. 5). In fact,

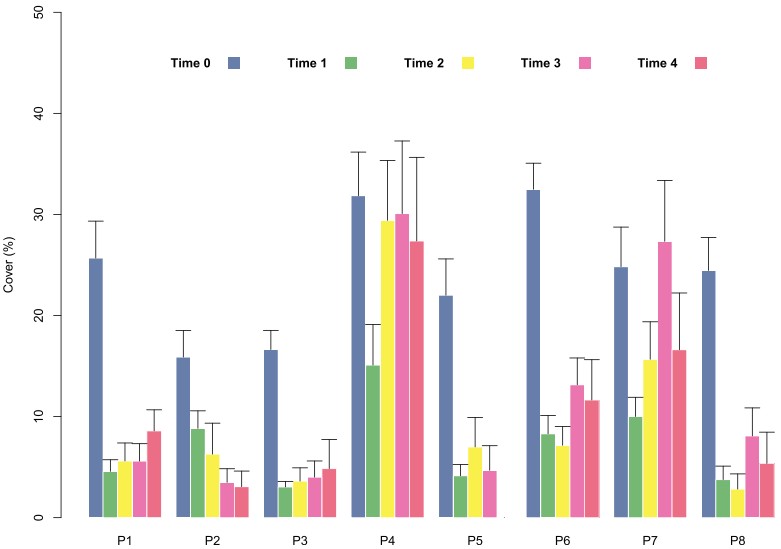

**Figure 3** Percent cover of *C. amentacea* (average + SE) on the clay tiles over time (Time 0–Time 4) in the eight patches at the receiving site.

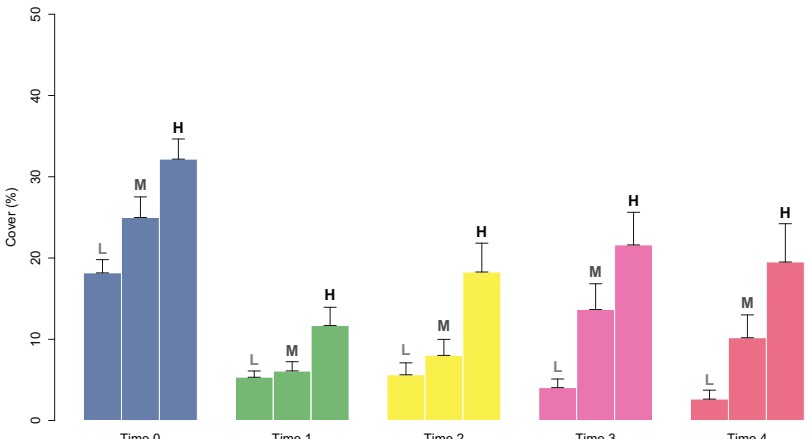

**Figure 4** Percent cover of *C. amentacea* juveniles (average + SE) in each cover class (Low-L, Medium-M, High-H) on the clay tiles over time (Time 0–Time 4) at the start of the experiment.

from Time 0 to Time 1, this percentage changed from 100% to 88% and was still over 40% after 2 months (Time 4). Similar to cover, survival also differed in the patches characterized by higher cover of juveniles at the start of the experiment. In fact, the GLM on the presence/absence of juveniles on the tiles at Time 4 using the cover class of each patch at the start of the experiment as a predictor variable showed significant differences among the classes ($p = 0.00161$). The Low class significantly differed from the others, while the Medium class did not differ from the High class (Table 1): this implies that cover at the start of the experiment $\geq 25\%$ is required for a good outplanting performance.

The growth of the ouplanted juveniles over time is shown in Figs. 6 and 7. The thallus length of the juveniles of *C. amentacea* was mostly under six mm at Time 0 ($3.22 \pm 0.08$, avg $\pm$ SE), Time 1 ($3.73 \pm 0.10$) and Time 2 ($4.67 \pm 0.13$). At Time 3 ($6.02 \pm 0.18$) and

**Table 1 Results of the GLMs of the differences in percent cover and presence/absence of juveniles of the cover classes (Low, Medium, High) at Time 4.**

| Presence/absence | | Estimate | Std. error | z-value | Pr(>\|z\|) |
|---|---|---|---|---|---|
| | High–Medium | 0.05084 | 0.28864 | 0.176 | 0.98288 |
| | **High–Low** | **−0.99550** | **0.36378** | **−2.373** | **0.01689** |
| | **Medium–Low** | **1.04634** | **0.32695** | **3.200** | **0.00387** |
| **Percent cover** | | | | | |
| | High–Medium | −0.8179 | 0.3683 | −2.220 | 0.0652 |
| | **High–Low** | **−2.2942** | **0.5732** | **−4.003** | **<0.001** |
| | **Medium–Low** | **1.4763** | **0.5642** | **2.617** | **0.0231** |

Note:
The significant differences are highlighted in bold.

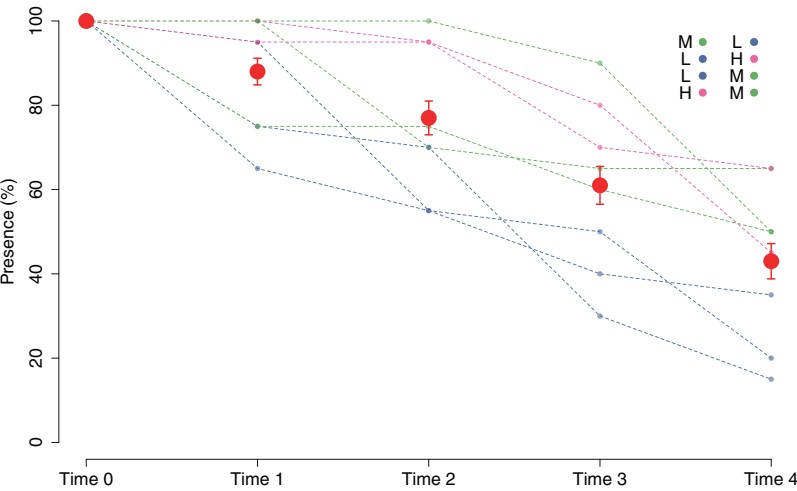

**Figure 5 Percentage of tiles with *C. amentacea* juveniles over time (Time 0–Time 4) according to presence/absence data for the individual tiles in the eight patches at the receiving site.** Eight patches were allocated to the different cover classes. Red dots represent average values across the patches (average ± SE). L, low cover class; M, medium cover class; H, high cover class.

Time 4, juveniles grew to up to 11 mm, with a higher number of individuals 8–10 mm in size at Time 4 (8.03 ± 0.22). After 6 months (Time 5), the outplanted juveniles reached three to six cm in length (last record, in February 2019).

## DISCUSSION

Outplanting represents an innovative technique for restocking of brown canopy-forming macroalgae (*Falace et al., 2018*), although its implementation consists of a set of delicate steps: (i) fertile material collection, (ii) culturing of juveniles in the lab, (iii) transport of juveniles to the field, and (iv) attachment of the juveniles to the rocky shore. In addition to the multiple set of phases, this technique needs to be adapted according to target species-specific requirements in the different phases of implementation.

The lab culture process involves several issues related to the wellness of the sensitive embryo stage. The thorough cleaning of the fertile apices before and after transport to the

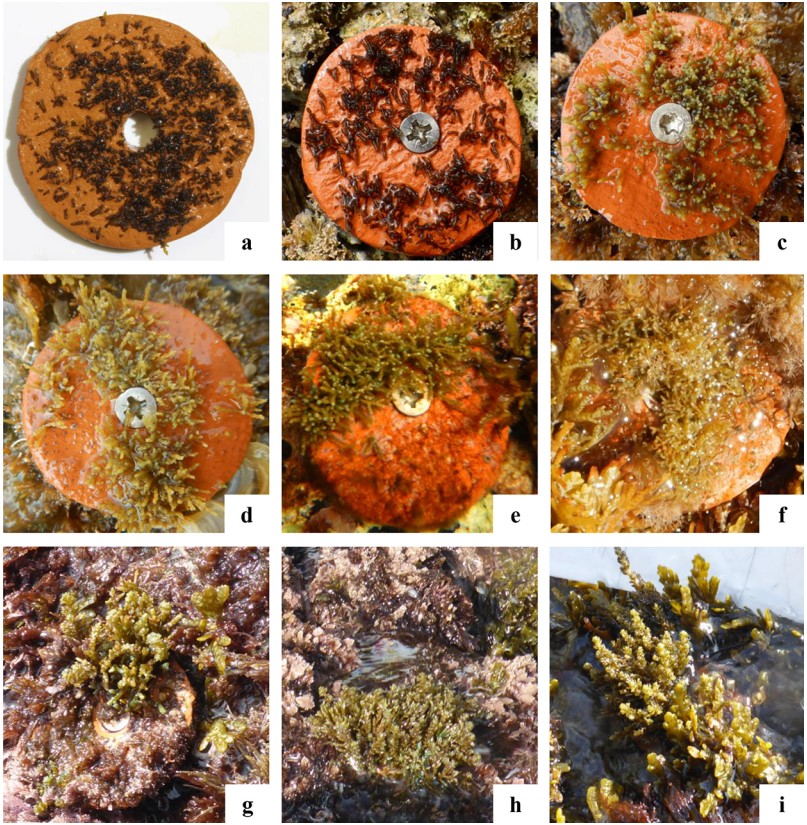

**Figure 6 Development of the outplanted juveniles on tiles over time (Time 0–Time 5) at the receiving site.** (A) Juveniles leaving the lab facilities (avg: 2.67 mm), (B) Time 0—outplanting day (avg: 3.22 mm), (C) Time 1—1 week (avg: 3.73 mm), (D) Time 2—2 weeks (avg: 4.67 mm), (E) Time 3—1 month (avg: 6.02 mm), (F) Time 4—2 months (avg: 8.03 mm), (G–I) juveniles 6 months after outplanting (three to six cm).

lab facilities to reduce epiphyte outbreaks and the presence of grazers helps to increase the chance of good performance. The *C. amentacea* embryos were cultured under their optimal temperature and light intensity conditions to maximize the yield (*Falace et al., 2018*), which can also unfortunately enhance spores, propagules, and bacteria proliferation. The implemented culture setting (e.g., the addition of antibacterial solutions and enhanced hydrodynamics) preserved the culture in good conditions, with healthy juveniles larger (2.65 mm) than those in previous *Cystoseira* restoration studies being obtained in only 3 weeks: *C. amentacea* (3 weeks—1.38 mm; *Falace et al., 2018*) and *C. barbata* (1 month 200–400 μm; *Verdura et al., 2018*). Furthermore, the length recorded 1 month after outplanting (6.02 mm; Fig. 6) is similar or even slightly larger than that recorded in a previous study on *C. amentacea* (4.73 mm; *Falace et al., 2018*) because the outplanting size was also larger in the present study.

The transport of early life stages from the nursery facility to the receiving site, which may be located a large distance away, as in this case (≈600 km), was found to not affect their fitness. The results show no significant differences in the percent cover of juveniles before and after transport. The cover of the juveniles on tiles at the time of arrival at the receiving site was even slightly greater than that when leaving the nursery facility

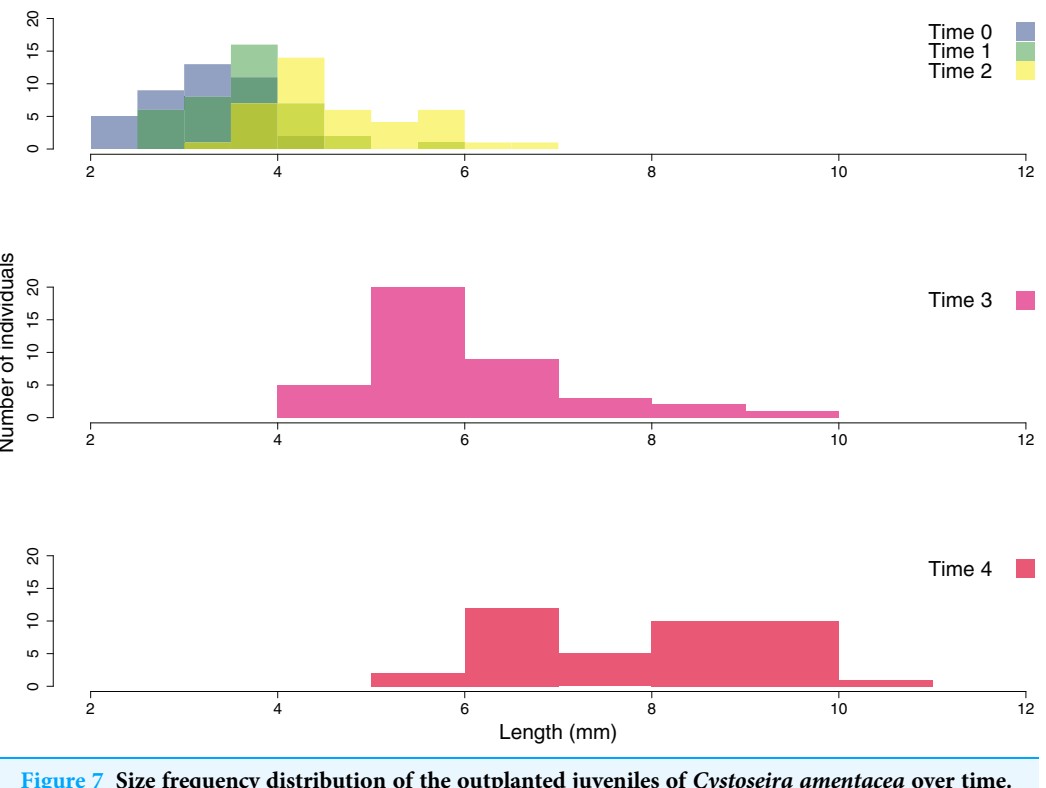

**Figure 7 Size frequency distribution of the outplanted juveniles of *Cystoseira amentacea* over time.**

(before transport—21.6%, after transport—24.2%; Fig. 2). This result was obtained maintaining a good temperature range (20–22 °C) during transport and during the attachment of the clay tiles in the field (outplanting action). Due to the high numbers, the clay tiles were transported, by rubber boat in cooled boxes to the rocky shore in several times over the course of the same day to prevent solar heat stress during attachment.

The attachment technique applied in this study, using pre-established screws instead of epoxy putty, which is generally used for both adults and juveniles (*Susini et al., 2007*; *Whitaker, Smith & Murray, 2010*; *Perkol-Finkel et al., 2012*; *Verdura et al., 2018*), increases the effectiveness of outplanting by reducing the time spent for attachment (the deployment of 400 tiles was performed in 5 h by six people) and the possible dislodgement by wave action on the day of deployment or the following days (*Susini et al. 2007*). This method additionally minimizes the esthetic and environmental impacts on the rocky shores and is reported to reinforce attachment, preventing the possible dislodgement of the clay tiles by wave action, as in the case of *Lessonia nigrescens* restoration (*Vásquez & Tala, 1995*). This is particularly important for midlittoral species living under high hydrodynamic conditions, as in the case of *C. amentacea*. In our study, we observed very positive results over the first 2 months (with the overall dislodgement of approximately 39 tiles). Unfortunately, the experiment was strongly affected by an unprecedented storm surge (winds over 170 km h$^{-1}$ and significant wave height of six m and maximum waves of 10 m height, over approximately 12 h). This storm surge caused the destruction of entire harbors and ships in marinas and caused great damage to the

shallow benthic communities. This event caused a large loss of tiles: 69 tiles survived the storm out of the 361 reported at Time 4. This means that, notwithstanding the exceptional strength of the storm, 20% survived the event, although only 16 of them hosted juveniles. Overall, the attachment method used in this study seems to be not only more environmentally friendly than the epoxy putty attachment method but also quite resistant to wave action. The screws themselves were not at all removed from the shore.

In addition to providing information for the optimization of culture and transport/outplanting techniques, our results stress the relevance of juvenile cover on the tiles. Relatively high cover ($\geq$ 25% of the tile covered at the time of deployment) ensures the survival of the outplanted juveniles (Table 1; Fig. 4). Midlittoral species are exposed to high solar heat and desiccation stress; therefore, a higher percentage of cover allows moisture and shading to be retained, enhancing the development of early juveniles (*Brawley & Johnson, 1991*; *Dudgeon & Petraitis, 2005*).

Another key issue regulating outplanting success is grazing, which was reported in previous studies, for example, on *Sargassum* species, where did not obviously affect the development of the outplanted juveniles in order to establish population in the restoration site, because of the low density of major grazers (seven individuals/0.25 m$^2$, *Yu et al., 2012*) and the use of cages to protect juveniles from grazing, until reaching a suitable grazing free size (*Yoon, Sun & Chung, 2014*). In this study, we did not formally test the effect of grazing, but we can exclude it having a relevant role. No grazers were observed crawling on the tiles at the time of sampling or in the photographs. In terms of small grazers (mostly small amphipods and isopods), their potential abundance may be estimated from the paper by *Thrush et al. (2011)*, which reports densities of grazing crustaceans in the range of nine individuals/20 cm$^2$ in the area, which is likely not high enough to exert a strong grazing effect. Furthermore, the tiles were located relatively high on the shore compared to the mean sea water level, and this could actually prevent grazing by fish and sea urchins, which are generally considered the most relevant herbivores (*Ling et al., 2015*; *Gianni et al., 2017*).

## CONCLUSIONS

Our findings show that outplanting midlittoral canopy-forming species is a feasible approach for restoration efforts. In particular, our study addressed the early steps of restoration and provides information on best practices for the outplanting phases of laboratory culture, transport, and reintroduction in the natural environment.

The laboratory culture phase requires appropriate species-specific protocols to reduce outbreaks of epiphytes, obtain high cover and a large size of the juveniles for field deployment, and increase the potential for restoration success. The use of an appropriate culture medium, the addition of an antibacterial solution and thorough fertile material cleaning are relevant elements likely to guarantee good culture performance and obtain high densities of healthy embryos.

The feasibility of large distance transport from the laboratory to the field has been remarkably shown, providing a potential option for replication also on a large scale.

The applied screw attachment technique was found to be effective in terms of increasing the efficiency of tile deployment and resistance to dislodgement, particularly in the early stages of restoration, when the traditional epoxy putty technique may be strongly affected by wave action, which is particularly relevant for midlittoral species.

These results are strongly encouraging for the implementation of restoration actions of canopy-forming species on a large scale, in light of EU guidelines.

## ACKNOWLEDGEMENTS

We would like to thank the contributions of all colleagues and students who helped in the field and during samples processing: Saul Ciriaco (Shoreline Soc. COOP), Sara Menon (Shoreline Soc. COOP), Marco Segarich (Shoreline Soc. COOP), Massimo Andreoli (Cinque Terre MPA), Enrico Agostini (Nemo-Italia), Davide Monteggia (University of Genoa), Maria Paola Ferranti (University of Genoa), Lorenzo Meroni (University of Genoa), Greta Fallanca (University of Genoa), Francesca Piga (University of Genoa). We are sincerely grateful to the referees Enric Ballesteros, Kiran Liversage and an anonymous third referee, as well as to the editor Anastazia Banaszak for improving this manuscript with their valuable comments.

### Funding
This study was supported by the LIFE financial instrument of the European Community, project ROC-POP-LIFE (LIFE16 NAT/IT/000816). The funders had no role in study design, data collection and analysis, decision to publish, or preparation of the manuscript.

### Grant Disclosures
The following grant information was disclosed by the authors:
LIFE financial instrument of the European Community, project ROC-POP-LIFE: LIFE16 NAT/IT/000816.

### Competing Interests
The authors declare that they have no competing interests.

### Author Contributions
- Gina De La Fuente conceived and designed the experiments, performed the experiments, analyzed the data, prepared figures and/or tables, authored or reviewed drafts of the paper, approved the final draft.
- Mariachiara Chiantore conceived and designed the experiments, performed the experiments, analyzed the data, contributed reagents/materials/analysis tools, prepared figures and/or tables, authored or reviewed drafts of the paper, approved the final draft.
- Valentina Asnaghi conceived and designed the experiments, performed the experiments, analyzed the data, authored or reviewed drafts of the paper, approved the final draft.
- Sara Kaleb conceived and designed the experiments, performed the experiments, approved the final draft.

- Annalisa Falace conceived and designed the experiments, performed the experiments, contributed reagents/materials/analysis tools, authored or reviewed drafts of the paper, approved the final draft.

### Field Study Permissions

The following information was supplied relating to field study approvals (i.e., approving body and any reference numbers):

Field experiments were performed under permission of the Cinque Terre Marine Protected Area.

### Data Availability

Percent cover and presence/absence data of the Cystoseira outplanted juveniles from T0 to T4 are available, respectively, in Data S1 and S2.

### Supplemental Information

Supplemental information for this article can be found online at http://dx.doi.org/10.7717/peerj.7290#supplemental-information.

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
