# Peer review of "First ex situ outplanting of the habitat-forming seaweed Cystoseira amentacea var. stricta from a restoration perspective"

_PeerJ, doi:10.7717/peerj.7290_

## Round 0.1 · original submission · Major Revisions

Three expert reviewers have evaluated your manuscript and their comments can be seen below and in a PDF. The reviewers agree that this is a novel study using juvenile algae in restoration, however there are a number of issues that have been pointed out.

There is a contradiction within the document as to the viability of the attachment technique that was employed. Many tiles detached in a storm, so this result would indicate that this method, though novel, has limitations. This needs to be pointed out and discussed in the document.

There are a number of informal or subjective comments that need to be substantiated with evidence or removed. For eample: it appears that no cultures were treated without antibiotics, which would serve as control cultures, therefore it is not correct to imply that the presence of antibiotics helped to maintain the cultures in good conditions.

It would be wise to conduct a thorough literature review and to ensure that any statements are correctly cited.

Finally, in general the writing style needs improvement particularlly ensuring a logical flow in the different sections as well as more precise language to avoid confusion. The abstract needs more details especially with respect to results; the introduction needs better flow; the methods need more details in particular on the grazing experiment and the size of the algae there were selected for the experiment; Results and discussion are confusing and need to be redone; the conclusions are too broad.

Once you have a final version, I strongly recommend that a native English speaker review the document.

·

Basic reporting

The paper by Gina de la Fuente and co-workers is interesting since they provide a methodology based on outplanting of juveniles of algae of the species Cystoseira amentacea in places that –according to ancient data- are able to thrive but that are currently absent. The study is novel in the sense that this species has never been subject to this kind of outplanting, i grows in exposed rocky shores, it is very vulnerable to nutrient excess and other anthropogenic factors such as urbanization, and the method of attachment had never been used.

English grammar is fine but sentences are sometimes confusing and can be improved in order to enhance comprehension by the reader. The introduction is well contextualized although authors should take care in be more precise when talking about canopy forming algae in general versus Cystoseira and of the whole ocen versus the Mediterranean. I had problems in understanding what they were talking about in some sentences. Ancient literature has to be revised more thoroughly since they usually only use the most recent papers but not previous studies dating back to before 2005 or to the last century.

As it has been said already, the manuscript can be improved for clarity. Figures and tables are fine.

Experimental design

Regarding the experimental design, it is very simple as they describe a methodology and they are interested in the partial and final outputs but not in testing the different factors they have used in the methodological design. For instance they use antibiotics in culture to improve survival of recruits and early juveniles but do not have a control situation (no antibiotics). Therefore they can not test if the use of antibiotics makes any improvement in the survival/growth of the early juveniles. In my opinion the study is just a description of the method used and not a test to slightly different methods. However, I do not complain about this unless the authors do not say in the discussion that antibiotics are very useful in improving survival

Validity of the findings

The research is well defined and the aims of the paper too. Methods -in my opinion- are described well enough to be replicated.

Data is robust (it could not be other way regarding the huge amount of tiles used in the study) and therefore results are sound.

Additional comments

The writing is confusing, mainly in the results and discussion chapter and the authors should try to improve this aspect. Conclusions are not well stated and authors should also try to give a clear message of the specific results they obtained, not to limit in general aspects of the study.

·

Basic reporting

The basic reporting is ok, except for the use of English language which needs moderate improvement, although the meaning is general clear.

A few other points: the abstract is too short and needs more exact details of what the results were. And the figure legends need more detail (the figures + legends should be mostly understandable without looking at the rest of the text).The use of paragraph structure in parts of the introduction is unusual, maybe the small paragraphs need to be combined, as some have only one sentence.

Experimental design

1) The research questions in the Introduction are strongly about restoration of intact functioning habitats, but these experiments are only applicable to very early stages of restoration (there is proper data only about the first 3 months). This experimental design does not thus allow much understanding of the possibility for eventual restoration success.

2) There are aims about assessment of grazers, but this part of the project is not properly developed. For example, it is not stated in the text where or when the grazers were collected from. It seems they were not collected from on top of the tiles (line 190), but I think sampling of grazers directly on the tiles would be the only way to get meaningful information. A proper hypothesis and statistical test would be ideal, e.g. comparison of grazer densities on vs off the transplanted algae.

3) Good data about size of the plants would be very useful for a project like this. The authors have some mention about measuring size of algae individuals (line 133) but again this part of the project is not well developed. In the results there is just a brief mention of sizes (line 186) with almost no details of how these measurements were made. If proper formal measurements were made, these should be displayed as a graph rather than just a series of informal photographs (Fig. 6) which are provided currently.

4) I do not think that the data about cover of algae from before to after transport to the field site will be very useful to most readers, and could be briefly mentioned in one or two sentences, or put in supplementary information.

5) The use of "cover classes" (line 152-155) is not ideal, as it seems three arbitrary categories have been chosen and the data put into them. These data are really continuous, not categorical data, so a regression analysis would be best. At the very least, the reasons for making those specific categories would need to be explained.

6) Overall, there was only limited use of statistics on the data. Analyses would be useful to see if the changes occurring over time were significant, whether the changes varied according to the different sites, correlations with grazer abundance etc. Only two tests are described (line 145-155), and the first of these (about changes before and after transportation) I do not think is very relevant (see point above).

Validity of the findings

1) As "most of the tiles were detached" (line 131) during a storm after only 3 months, there needs to be strong emphasis on how the experimental attachment method was actually not successful. There is text which makes it seem like the attachment method worked (e.g. line 253). Most tiles detaching after 3 months I do not think could be considered successful, regardless of whether the storm that happened was "unprecedented" (line 132). Transplantation substrata must remain for many more months/years if self-sustaining populations of macroalgae will establish.

2) Not only did the method for attaching the tiles not really seem to work, but the persistence of the algae also seemed to be mostly unsuccessful. After 3 months there was a 60% drop in the number of tiles that had the presence of any macroalgae (Fig. 5). If this trend continued, all tiles would have lost their macroalgae within a few more months. The fact that the presence of the algae is declining towards zero is not discussed. Overall, the only evidence of success is some photographs (Fig. 6) but no valid data is provided to show that the scene in the photographs is representative to what occurred on average.

3) The discussion text about grazers (line 238) is not valid unless more extensive and clearly explained data on grazers are collected.

Reviewer 3 ·

Basic reporting

In general the MS is relatively easy to read but there are some specific sentences where the flow and/or the English language should be improved to ensure that an international audience can easily understand the text. Eg:
- L40, Cystoseira is discussed, but the following line 41 the author refers back to all macroalgae. The author would improve the logical flow of the paper by revising the structure of this paragraph so that it reads large>small scale.
- In terms of restoration techniques (L54), transplanting adults does enhance recruitment potential. I recommend the author groups the first two techniques together or ranks them i.e 1.(i) technique as (ii) technique b compared to 2. Technique c
I suggest the author revises the reporting here by condensing sentences and having a native English language speaker review your text before next submission.
The author has also included some (not all) of the relevant literature for their topic, however in some cases statements made in the paper were not referenced correctly or sufficiently. E.g L30-31: “Around thirty-five percent of brown algae species (Fucales and Laminariales; Guiry and Guiry, 2019) contribute to the structure of the coastal landscapes, playing a key role as ecosystem engineers, providing an important value as natural capital.”
Several points are made in this sentence, however only the fact that “35% of brown algal species… is referenced. I suggest the author revises where they have placed the referencing and includes additional references where none are given.
The MS would benefit from expansion of many of the points made so that they include specific examples (and referencing) relevant to the local context. Some examples from the Introduction include:
- No information is given about when, or why C. amentacea disappeared. Could the authors elaborate more on this in order to justify why they believe restoration is an appropriate measure, given that the population has been replaced by a more tolerant species and they have not mentioned that any of the factors contributing to the decline of these populations have been removed.
- Could the author elaborate on the use of in situ vs ex-situ techniques and their relationship to dispersal capacity? (L64). Many kelps are thought to have dispersal limited to <10 metres from the parent alga – is this high dispersal capacity compared to C. amantacea?
Reporting of the methods results is also vague and incomplete in many places (specific cases in Methods section are discussed in the Experimental design section below), while some unnecessary information was given instead – eg:
L118 Dates of transplanting and the material of the boat used to transport the tiles are not relevant, but please include temperature and condition information.
L166 While the reader may look at the figure, the author should be reporting the results (eg, mean and s.e) in the Results section, not just describing trends.
The author should also report the overall presence/absence of the plots following the storm ie at time 5 as rough weather is expected in any field-based restoration attempt.

Experimental design

The authors here report some promising results from the first attempt at restoring Cystoceira amentacea, however they do not explicitly test everything stated in the abstract. A clearer outline of the aims/objectives for this study or including additional data and analyses comparing different methods of transport and grazing could improve this.
There is also an urgent need to include more specific information in the methods - both to improve the reader’s understanding of what was tested and to make it reproducible. Some examples of where more information is necessary include:
- Recording the GPS points of the sites (L81)
- The size of the area that they collected algae from (L81)
- The temperature and conditions of the “environmentally controlled rooms” (L91) you were using in the laboratory
- The temperature and photoperiod used to culture (L100) (I appreciate this paper builds on Falace et al 2018, however it should include the optimal culture conditions here as a stand-alone document)
- The statistical packages (and versions) used for data analysis.
Some additional data and analyses could also be presented here without much effort on the author’s behalf, including height/growth measurements of the outplanted juveniles and whether there was a difference between patches.
It is interesting that the author mentions that grazing is one of the predicted barriers to restoration success, and I commend them on trying to associate grazer abundance to restoration success. However, could the author please elaborate on why they collected abundance data from infauna as opposed to other mobile grazers which are often the cause of overgrazing in seaweed populations? Secondly, I suggest that the author includes several more sentences describing how they assessed grazing in the methods as it is not immediately clear how they came to the conclusion that “Grazing did not clearly affect the outplanted juveniles”L189. The authors have valuable data on benthic grazer abundance which they could do more with to support this claim, for example by looking at the relationship between grazer abundance and loss in percent cover in each plot.

Validity of the findings

The general overview of findings presented in the results and discussion includes some elements of subjectivity and also informal comparison to other studies. I suggest that the author moves any value judgements and assessments to the discussion section, but they should also take care not to over-estimate the conclusions that can be made from this study alone. Eg: L204-205: “The new culture medium formula, the antibacterial mixed solution and the enhanced hydrodynamics preserved the culture in good conditions” This was not tested with the study design, therefore I would encourage the author to either adjust this sentence or include experimental data supporting this conclusion in the study.
Some of the conclusions made here are also confusingly reported or contradictory eg: L217 the author discusses cover of juveniles on tiles having increased, but in the results it says the overall number of tiles with juveniles had decreased. If you are to include this result in the discussion, please give more information in the methods and results.
L229 it says that the methods used here “reinforces...the attachment avoiding the possible dislodgement of the clay tiles by wave action" however in the beginning it stated a storm severely affected the tiles...please explain.

Additional comments

The aim of this paper is to report the first ex situ outplanting of the habitat forming seaweed C. amentacea for restoration purposes. The paper describes the methods used to collect wild specimens of C. amentacea, culture juveniles in the lab under pre-defined conditions and provides some estimates of survival and growth over the first 4 months following outplanting in the field.
This is a novel study with some interesting results regarding growth and survival of outplanted C. amentacea in the field, however there are several issues with basic reporting and analysis that are fundamental to the understanding and conclusions drawn in the paper and need to be improved before this MS should be considered for publication. I suggest that the authors revise and resubmit this MS and have included some examples where it could be improved.

---

## Round 0.2 · Minor Revisions

A final evaluation of your manuscript has been received and the reviewer is satisfied with the changes that have been made to the manuscript. There are just a few edits that need to be made as can be seen in the attached PDF (see lines 159, 205, 262 and 377). Once you have made these changes and uploaded the new version of the manuscript, I will recommend that it be accepted.

·

Basic reporting

Authors have improved the manuscript a lot according to my suggestions, In my opinion it is now, with very minor changes (see the pdf file), sitable for publication

Experimental design

ok

Validity of the findings

ok

Additional comments

ok

---

## Round 0.3 · accepted · Accept

I am satisfied with the changes made to the manuscript.